# Molecular Epidemiology of Rabies in Wild Canidae in Tunisia

**DOI:** 10.3390/v13122473

**Published:** 2021-12-10

**Authors:** Zied Bouslama, Habib Kharmachi, Nourhene Basdouri, Jihen Ben Salem, Samia Ben Maiez, Mariem Handous, Mohamed Saadi, Abdeljalil Ghram, Imed Turki

**Affiliations:** 1Laboratory for Rabies Diagnostics, Institute Pasteur of Tunis, Belvedere, Tunis 1002, Tunisia; h.kharmachi@yahoo.fr (H.K.); nourhene.basdouri@yahoo.com (N.B.); jihen.bensalem@pasteur.tn (J.B.S.); samia.benmaiez@pasteur.tn (S.B.M.); mariem.handous@pasteur.tn (M.H.); mohamed.saadi@pasteur.tn (M.S.); 2Faculty of Sciences, Université Tunis El Manar, Tunis 2092, Tunisia; 3Laboratory of Epidemiology and Veterinary Microbiology, LR 16 IPT 03, Institut Pasteur de Tunis, Université Tunis El Manar, Tunis 2092, Tunisia; abdeljalil.ghram@pasteur.tn; 4Service des Maladies Contagieuses, Ecole Nationale de Médecine Vétérinaire-Sidi Thabet, Université Manouba, Sidi Thabet 2020, Tunisia; turkiimed58@yahoo.fr

**Keywords:** rabies, wild canidae, Tunisia, phylogeny

## Abstract

Rabies is a viral zoonosis that is transmissible to humans via domestic and wild animals. There are two epidemiological cycles for rabies, the urban and the sylvatic cycles. In an attempt to study the epidemiological role of wild canidae in rabies transmission, the present study aimed to analyze the genetic characteristics of virus isolates and confirm prior suggestions that rabies is maintained through a dog reservoir in Tunisia. Virus strains isolated from wild canidae were subject to viral sequencing, and Bayesian phylogenetic analysis was performed using Beast2 software. Essentially, the virus strains isolated from wild canidae belonged to the Africa-1 clade, which clearly diverges from fox-related strains. Our study also demonstrated that genetic characteristics of the virus isolates were not as distinct as could be expected if a wild reservoir had already existed. On the contrary, the geographic landscape is responsible for the genetic diversity of the virus. The landscape itself could have also acted as a natural barrier to the spread of the virus.

## 1. Introduction

The rabies virus belongs to the *Lyssavirus* genus. It is a negative-sense single-stranded RNA virus with a genome size of approximately 12 kb. It contains five genes, namely the nucleoprotein (N), the phosphoprotein (P), the matrix protein (M), the glycoprotein (G) and the large protein or polymerase (L), and a non-coding region [1,2]. Among these, the nucleoprotein (N) is the most conserved gene, and the phosphoprotein (P) and glycoprotein (G) are the most variable ones [3]. To date, 17 different viral species have been identified [4]. Rabies is a viral infectious zoonotic disease endemic in many regions of the world, particularly Asia, Africa, and South America. In Tunisia, rabies is enzootic and endemic. Rabies virus (RABV) can be associated with a variety of mammalian hosts that maintain independent epidemiological cycles within geographical locations, resulting in the differentiation of the virus into several lineages [5,6]. RABV strains that circulate in dogs (*Canis lupus familiaris*) are responsible for more than 99% of all human cases worldwide [7,8]. However, different wild mammals have been identified as RABV reservoirs, such as bat-eared foxes, black-backed jackals, and yellow mongooses in Southern Africa and red foxes in Europe and the Middle East. In different parts of the world, these species maintain sylvatic transmission cycles that can be independent from the primary urban cycle in which the dog is the main reservoir [9,10].

Rabies viruses isolated from foxes in Europe belong to the same cosmopolitan lineage associated with dogs. However, it has unique genetic characteristics and thus forms a separate clade [11]. In North Africa, and particularly in Tunisia, the molecular characterization of rabies virus isolated from dogs [12], domestic animals [13] and humans [14] allowed for the sole identification of canine rabies clade Africa 1-a, belonging to the cosmopolitan lineage, showing that dogs are the main reservoir species in the area [12,15]. However, no study has yet been conducted on rabies viruses isolated from wild animals in Tunisia to investigate the existence of a sylvatic circle. The laboratory for rabies diagnostics at Institut Pasteur of Tunis (IPT) is the national reference laboratory and the only one authorized for rabies diagnostics in Tunisia. Yearly, an increasing number of samples are tested, exceeding 1000 samples per year since 2014. According to the national commission of rabies control, samples submitted to the laboratory are mostly from dogs identified as suspected cases, while a maximum of three samples per year are collected from wildlife [16]. A suspected case is identified as an animal that died after presenting neurological symptoms or after having bitten a human. In turn, among animal rabies cases reported yearly in Tunisia, dogs represent 60%, while wild animals only represent 0.3 % [15]. For example, among 383 rabies cases reported in Tunisia in 2018, 236 cases were identified in dogs and 3 were human cases. It is known that a certain number of cases can be recorded in wild canids even when the country adheres to mass dog vaccination. Indeed, rabies cases are broadly recorded in wild canids among North Africa, including in Tunisia, even if these host species are poorly studied for their potential role in rabies transmission. On the contrary, studies performed in Middle Eastern countries such as Iraq, Jordan, and Syria confirmed the coexistence of urban and sylvatic rabies cycles [17]. The objective of this study was the molecular characterization of rabies viruses isolated in Tunisia from wild canids, specifically foxes and jackals, in order to compare them with variants that were previously reported from dogs in the area. In addition, we also analyzed and discussed the temporal evolution of variants in comparison with North African isolates.

## 2. Materials and Methods

### 2.1. Origin of Virus Isolates

Animals suspected to be dead from rabies infection were collected by the veterinary services and sent to the IPT laboratory to undergo necropsy, brain extraction and rabies testing using a FAT test, as recommended by the OIE and the World Health Organization (WHO) [18,19]. All samples that tested positive for rabies were stored in the rabies laboratory biobank. In total, we selected 11 brain samples collected from wild animals, namely five foxes and six jackals, and 14 brain samples isolated from domestic dogs in the same location and during the same period of time (same delegation, same month), as shown in Table 1. These samples were characterized and integrated into the phylogenetic analysis together with additional sequences downloaded from GenBank.

### 2.2. RNA Extraction and Amplification

RNA was extracted using TRIzol reagent (invitrogen^®^) as described in previous studies [20]. Brain samples were homogenized in 1 mL of TRIzol reagent and 0.2 mL of chloroform. After centrifugation (11,000 rpm during 15 min), the supernatant was mixed with 0.5ml of isopropanol. After a second centrifugation (11,000 rpm during 10 min), the collected supernatant was discarded, and the pellet was washed using 1 mL of ethanol. The tubes were then dried and eluted in 50 µL of RNase free water. In the case of wild canids, we characterized the partial Nucleoprotein gene with a hemi-nested PCR protocol using the SuperScript III One step RT-PCR kit (invitrogen) for the first step and the platinum Taq (Invitrogen) in the second step, following the manufacturer’s instructions. A partial sequence of the phosphoprotein gene was characterized for both wild canids and domestic dogs with in one step using an Affinity Script One-Step RT-PCR kit (Agilent) as recommended by the manufacturer. The primers used for both analyses are presented in Table 2.

### 2.3. DNA Sequencing

PCR products were purified with ExoSAP and sequenced using the BigDye Terminator v3.1 cycle sequencing kit (Applied Biosystems, Waltham, MA, USA). Each amplicon was sequenced using one forward and one reverse primer to determine consensus sequences reaching up to 751 bp (N gene) and 1017 bp (P gene), using an ABI PRISM 3500 Genetic Analyzer (Applied Biosystems). Original sequences were trimmed and edited using BioEdit software. Then, they were aligned along with a set of sequences downloaded from GenBank using MAFFT online [22]. All sequences were deposited in GenBank (accession numbers OK275663-OK275695).

### 2.4. Maximum Likelihood Phylogeny

In order to determine taxonomy and clade membership for each virus isolate, we performed maximum likelihood (ML) phylogenetic analysis on the nucleoprotein dataset [23]. We first determined the best-fit substitution model by Smart Model Selection (SMS) [24], and certain GTR model + Gamma 4 categories were selected as the best parameters based on the lower Akaike Information Criterion (AIC). We then implemented ML phylogeny in PhyML with 1000 bootstrap replicates, using PhyML online software available at “http://www.atgc-montpellier.fr/phyml/, accessed on 26 January 2021”. The resulting tree was edited and annotated on FigTree 1.7 [25].

### 2.5. Bayesian Phylogeny

To determine the time to common ancestry among RABV clades circulating in Tunisia, we conducted a Bayesian MCMC phylogenetic analysis on the phosphoprotein dataset [26]. Initial substitution rates were estimated for use as tree priors by using Tempest [27]. A substitution model was averaged by using the bModelTest Package in Beast2 [28]. A Bayesian phylogenetic tree was constructed in Beast2 using a TN93 model with a discrete Gamma distribution among variations and a relaxed molecular clock model [29]. A Bayesian skyline plot model was specified as a tree prior. The analysis was set to run for 30 million iterations. ESS values were examined using Tracer software. A Maximum Clade Credibility (MCC) tree was constructed after discarding the first 10% burn in using Tree Annotator V2.6.2. Rates of nucleotide substitution and the time to most recent common ancestor (tMRCA) were estimated. Time tree was edited and visualized using iTOL online software available at: “https://itol.embl.de/, accessed on 18 March 2021”. The spatial distribution of variants was edited using the ggmap package v3.0.0 [30].

### 2.6. Reservoir Species Analysis

A pairwise genetic distance matrix was calculated using the ‘Ape’ package on R software and then visualized using the ‘ggplot2′ package on R software. The dataset included P sequences obtained from wild canids and dogs sharing similar sampling dates and locations. To identify the most likely animal species reservoir for each Clade, a second analysis was carried out on Beast2 using the animal host species as a discrete trait, as specified in a previous study [16]. The analysis was set to run for 30 million MCMC iterations. An MCC tree with traits was set and annotated using TreeAnnotatorV2.6.2, then visualized and edited in Fig-tree 1.7.

## 3. Results

### 3.1. Virus Taxonomy and Clade Membership

After trimming and alignment, the obtained N gene sequences were up to 481 bp in length. Original N sequences were obtained for six jackals and five foxes. A maximum likelihood inferred tree was divided into six different lineages, identified as Asian, Cosmopolitan, Africa-2, Africa-3, Africa-4, and Arctic-related lineages (Figure 1), all of which are of the RABV virus species. Cosmopolitan lineage was divided into four clades, which were Africa-1a, Africa-1b, Africa-1c, and European Fox-related strains (Figure 1). All Tunisian viruses sequenced in the present study clustered with the clade Africa-1a within the Cosmopolitan lineage, together with the canine-related African rabies strains.

### 3.2. Comparison of Wild Canidae Virus Isolates with Tunisian Canine Variant Strains

We compared viruses associated with foxes and jackals in Tunisia with variants previously identified in Tunisian dogs, which were downloaded from GenBank. The resulting ML tree is displayed in Figure 2, showing that variants found in Tunisian wild canids cluster together with those identified in dogs. The phylogenetic tree was further divided in two sub-clades that were independent from the host species but related to the area of sampling, identified as northwest (NW) and northeast–center–south (NCS). All viruses isolated from wild canids clustered with high support within the northeast–center–south (NCS) sub-clade, except for the isolate 8112, associated with a jackal found in 2006 in the El Kef governorate (northwest of Tunisia). Another interesting result was that isolate 10582, associated with a jackal that originated from the Kasserine governorate (center west) in 2011, was mostly related with the isolate DogTN/Ks98 (isolated from Kasserine in 1998). Although they belonged to the NCS sub-clade, they did slightly deviate from other NCS isolates. This deviation was supported by a node with high bootstrap values (100%). The old Tunisian isolates, downloaded from GenBank (Tunisia 1986), also belonged to the NW sub-clade.

### 3.3. Reservoir Species Analysis

To better understand virus transmission between dogs and wild canids, we performed a pairwise genetic distance analysis followed by a Bayesian analysis using a dataset composed of sequences of the phosphoprotein gene isolated from wild canids and dogs that shared the same sampling times and locations. Original P sequences were obtained for six jackals, three foxes and fourteen dogs. A pairwise distance matrix, calculated and visualized on R software, showed 100% similarity (0% dissimilarity) between viral sequences associated with animals that shared the same sampling location, irrespective of the animal species (Figure 3). Similarly, the MCC tree built using the host species as a discrete trait (Figure 4) confirmed that variants sharing similar sampling dates and locations cluster together independently of the animal species. The origin (ancestry) of most of the clusters defined within the tree was represented by a canine isolate.

### 3.4. Time Scale of Rabies Virus Evolution in Tunisia

In order to estimate the time scale of divergence of the Tunisian rabies variants, we performed a Bayesian analysis using the phosphoprotein dataset, including sequences from the study, plus two sequences from Algeria, and eight additional sequences from Tunisia (Table 3) that were downloaded from GenBank. Molecular clock analyses implemented in Tempest confirmed a positive correlation between time and substitution events and estimated a slope rate of 5.4 × 10^−4^ substitution/site/year that was thus set as a prior for the mean clock rate. The MCC tree resulting from evolutionary analyses is displayed in Figure 5. The overall tree structure confirms that Tunisian isolates are divided into two phylogenetic groups, NW and NCS, that diverged a few decades ago; the most recent common ancestor (tMRCA) between these sub-clades was estimated to occur in the year 1957 (Figure 5). The NCS variants are further divided in two clusters, NCS-1 and NCS-2. RABV variants belonging to the cluster NCS-1 are located in the northeast, the center-east and the southeast of the country, while cluster NCS-2 includes viruses found in the northwest and the center-west (Figure 6). Interestingly, the two Algerian variants included in this study clustered within the cluster NCS-2. The time of the most recent common ancestor (tMRCA) between NCS’s clusters was estimated to occur in the year 1989 (95% HPD: 1980–2000), around thirty years after the divergence between the two sub-clades. The virus isolate number 8112, associated with a Jackal in El Kef in 2006, was highly divergent from all other sequences of the P gene alignment, although it clustered with NW variant based on phylogenetic analyses performed on the N gene. Its most recent common ancestor was estimated to occur in the year 1892.

## 4. Discussion

Our study provided a genetic characterization of rabies viruses isolated from wild canids in Tunisia and broadly in North Africa, using a stepwise approach, including viral isolation, sequencing, and phylodynamics reconstruction. The study highlighted interesting spatial and temporal dynamics of rabies virus evolution that, in Tunisia, are independent of the host species. Phylogenetic analysis of the nucleoprotein sequences confirmed that all Tunisian isolates originated from wild canids belong to the Africa-1a clade of the Cosmopolitan lineage, commonly associated with dogs in several African localizations, especially in the north and the east [3,12,32]. These isolates do not form a distinct cluster within the clade but are mostly related to dog sequences. The inclusion of animal species as a discrete trait in the phylogenetic analysis suggests that all clusters defined in Tunisian canines and wild canids have a rabies virus from domestic dogs as an ancestor. These results differ from what is known for the clade associated with red foxes (*Vulpes vulpes*) in Europe that, despite belonging to the same cosmopolitan lineage, show unique characteristics likely resulting from the adaptation to this wild host [9] and clearly diverge from the root of all North African variants [33]. Each viral variant has its specificity for a host reservoir encrypted in its genetic settings [11,31]. Our results also suggest that there is not any independent spillover of rabies virus in wild canids in Tunisia. This is not the case for the neighboring Middle Eastern region (Iran, Irak, Syria, Jordan) where both urban and wild rabies cycles exist in the same country [16]. When aligning the Tunisian wild canids sequences with those of the Tunisian dog strains downloaded from GenBank, the ML phylogeny showed that all isolates that have similar sampling dates and locations strongly cluster together independently of the animal species. Specifically, the current genetic diversity of the Tunisian isolates is represented by two major and geographically distinct phylogenetic sub-clades, previously defined by Amouri et al. (1986) as north–east–center–south (NCS) and northwest (NW) [12]. Our analyses showed that very old Tunisian isolates (Tunisia 1986) and two Algerian sequences (ALG/08-153 and ALG/08-200) clustered together with the Tunisian isolates. In addition, the distribution of Tunisian isolates (Figure 6) showed two clusters within the NCS sub-clade that are spatially distinct. Variants from the newly identified cluster, NCS2, are exclusively located in the western parts of the country, precisely the northwest and the center-west. Interestingly, this cluster also included variants isolated from the northeast of Algeria (close to the northwest of Tunisia), suggesting a possible epidemiological link between these neighboring countries. Previous studies that analyzed rabies strains in Algeria, Morocco, and Tunisia suggested the grouping of viruses according to their country of origin [17]. On the other hand, we confirmed the genetic distinction of two sub-clades and further genetic clusters that strictly associated with different geographical areas and are evolving independently. Our data are consistent with results from a preliminary investigation undertaken on dog rabies cases in Tunisia (2011–2016) and support distinct epidemiological cycles of dog rabies in the northwest and northeast [34]. Viral variants associated with wild canids follow the same pattern seen for dogs, suggesting that the geography rather than the animal species is responsible for the phylogenetic structure. In addition, wild canids located in the governorate of Kasserine (NCS) can be infected with variants from both clusters NCS1 and NCS2, according to their city of origin. In particular, we detected variants from cluster NCS-1 in Sbeitla and El Hammar and from NCS-2 in Thala. Thala is geographically separated from Sbeitla and El Hammar by the presence of Chaambi and Semmama Mountains (Figure 6), supporting the fact that geographical barriers have a critical role in the evolution of rabies virus. Multiple studies have identified the role of natural landscape features such as mountain ranges and waterways as natural barriers to virus spread [35,36,37]. Specifically, elevation has been proven to act as a barrier to the dissemination of the rabies virus [35], given that human settlements, and therefore dog density, are less common at high elevations [38]. The Tunisian landscape is characterized by the presence of the dorsal ridge, the Tunisian part of the Atlas Mountains (including Jebal Chaambi and Jebal Semmama) that cross along the country from the center-west to the northeast. On both sides of the dorsal ridge, on the plains of the northwest and the center-east, scattered farms are widely spread. Domestic animal and dog presence is also influenced by such landscapes [39]. Thus, the dorsal ridge may act as a natural barrier, responsible for the geographical dispersion of both branches of the NCS variants. In addition to the latter hypothesis, if a spillover existed in wildlife, the viruses isolated from wild animals living on both sides of the same mountain would have had close genetic characteristics. On the contrary, mountain chains seem to act as a natural barriers to virus spread. 

In the present study, we performed molecular clock analysis using sequences of phosphoprotein but not nucleoprotein, because R squared analysis only showed positive correlation on the P dataset. In addition, the higher genetic variability of the phosphoprotein gene makes it a better choice for phylodynamics reconstruction [3]. In fact, the estimated substitution rate using Tempest and further confirmed by Tracer was 5 × 10^−4^ substitutions per site per year. This rate did not differ widely with previous studies of lyssavirus evolution [3,40]. The relaxed molecular clock analysis suggested that the diversification of the two sub-clades of Tunisian rabies variants (NCS/NW) occurred during the second half of the twentieth century at the latest. Genetic clusters within the NCS sub-clade diverged around 30 years ago; 95% HPD of the tMRCA estimates for the two nodes not overlapping. Additionally, the NCS-2 cluster is genetically related to both Algerian rabies isolates, probably introduced to Algeria before 1990. These age estimates are consistent with previous analyses of RABV in North Africa, where the most recent common ancestor for all North African RABV was estimated to have existed during the period 1878–1945 [17]. Thus, the diversification of the two Tunisian sub-clades (NCS/NW) is more recent than that of all the North African clades. Our main limitation was the inability to access recent North African sequences in GenBank. This might have affected our accuracy in tMRCA estimation. In addition, because of the unavailability of recent Algerian sequences from the northeast (near the Tunisian border), it was not possible to reliably distinguish whether the virus isolates were the result of introduction of the Tunisian variants into Algeria.

## 5. Conclusions

Throughout this work, we demonstrated that rabies viruses isolated from dogs and wild canids are not genetically distinct but rather evolve according to the geographical area. Viruses from both sides of the dorsal ridge of the Atlas Mountains have evolved independently, most likely through an urban cycle maintained by dogs. It can therefore be recommended to the veterinary services that the fight against rabies in Tunisia remains dependent on fighting dog-mediated rabies. Vaccination campaigns focusing on dogs should be maintained and undertaken in coordination with laboratory-based surveillance, dog population management, vaccination awareness and post-exposure prophylaxis.

## Figures and Tables

**Figure 1 viruses-13-02473-f001:**
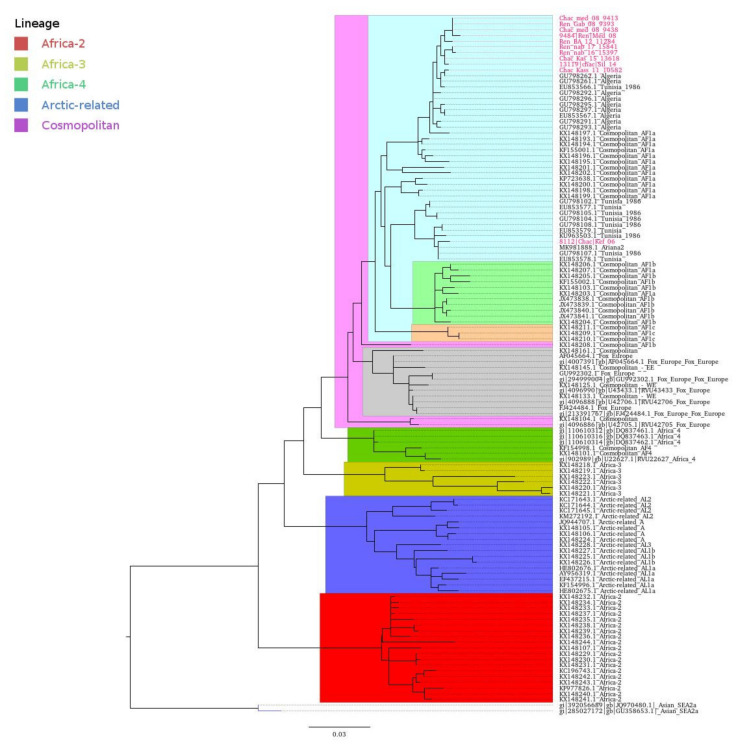
Phylogenetic analysis of rabies virus isolated from wild canidae in Tunisia. Cosmopolitan lineage (highlighted in purple) comprises Africa-1a (light blue), Africa-1b (light green), Africa-1c (orange), and fox-related strains (grey).

**Figure 2 viruses-13-02473-f002:**
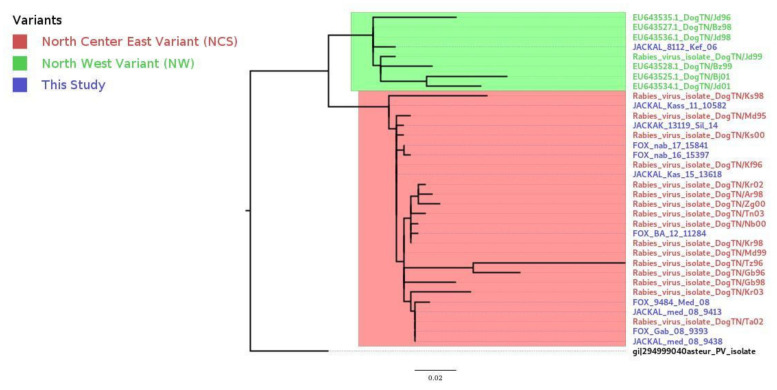
Phylogenetic analysis of rabies virus isolates in Tunisia.

**Figure 3 viruses-13-02473-f003:**
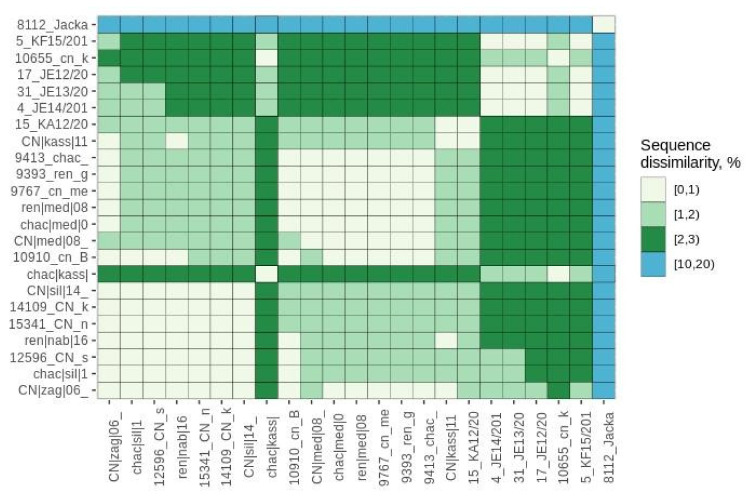
Heatmap representing the pairwise genetic distance between viral sequences associated with wild and domestic animals.

**Figure 4 viruses-13-02473-f004:**
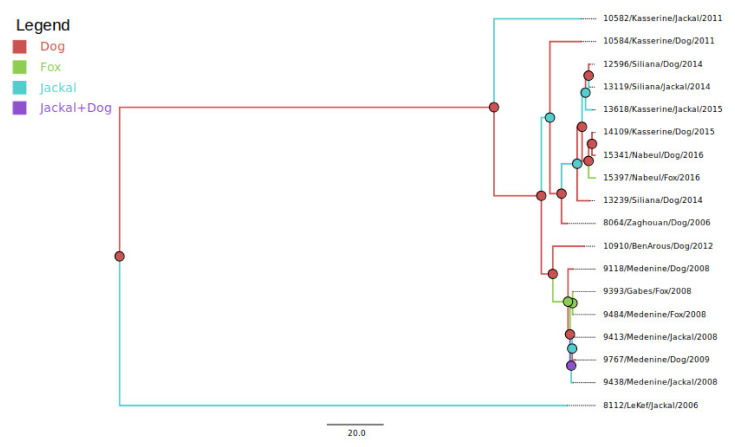
Species tree with traits representing the animal species probably associated with the ancestral node.

**Figure 5 viruses-13-02473-f005:**
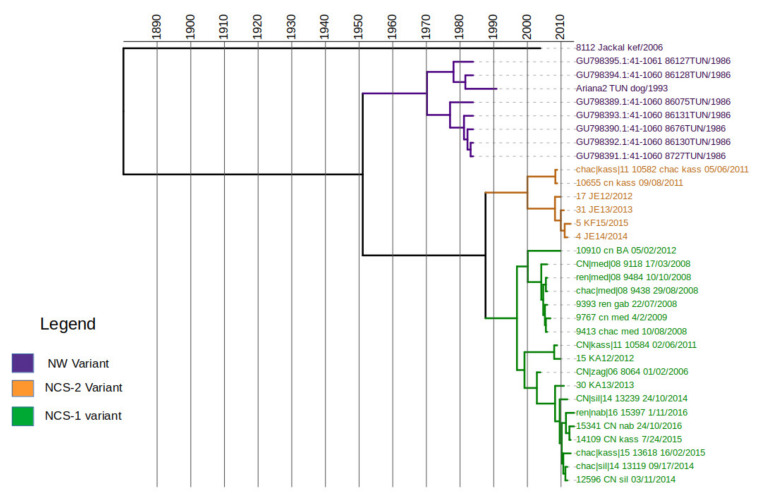
Time scale divergence of the Tunisian rabies phylogenetic variants.

**Figure 6 viruses-13-02473-f006:**
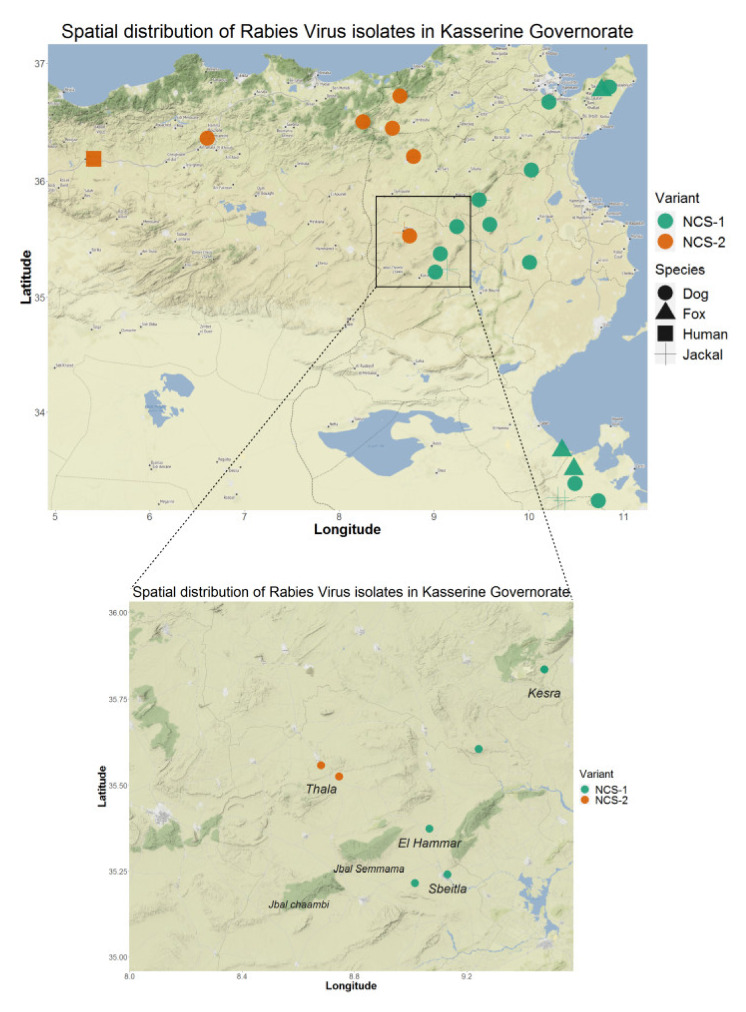
Spatial distribution of branches of the NCS variants.

**Table 1 viruses-13-02473-t001:** List of virus isolates selected for molecular characterization. Information on location and sampling date, corresponding to the animal’s death, were extracted from the laboratory’s database.

Isolate	Animal Species	Collection Date	Location (Governorate)	Location (Delegation)	Latitude	Longitude	Nucleoprotein Sequence Accession Number	Phosphoprotein Sequence Accession Number
15841	Fox	10 April 2017	Nabeul	Nabeul	36.4911995	10.6626997	OK275685	-
15397	Fox	1 November 2016	Nabeul	El Mida	36.773047	10.7656	OK27568	OK27568
11284	Fox	17 November 2012	Ben Arous	Mhamdia	36.647136	10.066214	OK275688	-
9484	Fox	10 October 2008	Medenine	Sidi Makhlouf	33.499538	10.473707	OK27569	OK275673
9393	Fox	22 July 2008	Gabes	Mareth	33.663899	10.348131	OK275692	OK275670
9413	Jackal	10 August 2008	Medenine	Beni Khdache	33.436471	10.206275	OK275691	OK275671
13618	Jackal	16 February 2015	Kasserine	Sbeitla	35.241052	9.131288	OK275687	OK275681
13119	Jackal	17 September 2014	Siliana	Kesra	35.8359985	9.4758101	OK275693	OK275679
10582	Jackal	5 June 2011	Kasserine	Thala	35.557643	8.680733	OK275689	OK275675
9438	Jackal	29 August 2008	Medenine	Beni Khdache	33.436471	10.206275	OK275690	OK275672
8112	Jackal	23 February 2006	El Kef	Dahmani	36.021820	8.906841	OK275695	OK275668
12596	Dog	3 November 2014	Siliana	Kesra	35.8359985	9.4758101	-	OK275678
14109	Dog	24 July 2015	Kasserine	Sbeitla	35.2153015	9.0153503	-	OK275682
10910	Dog	2 May 2012	Ben Arous	Fouchana	36.6683998	10.2124996	-	OK275677
15341	Dog	24 October 2016	Nabeul	Menzel Temime	36.796965	10.853034	-	OK275683
13239	Dog	24 October 2014	Siliana	Rouhia	35.7809982	9.0813599	-	OK275680
10584	Dog	2 June 2011	Kasserine	El Hammar	35.3737984	9.0671101	-	OK275676
9118	Dog	17 Mars 2008	Medenine	Ben Guerdane	33.2186012	10.7330999	-	OK275669
8064	Dog	1 November 2006	Zaghouan	Nadhour	36.0915985	10.0277004	-	OK275667
9767	Dog	2 April 2009	Medenine	Medenine Nord	33.3704987	10.4882002	-	OK275674
10792	Dog	31 January 2011	Jendouba	Ouled Mliz	36.4488983	8.5616598	-	OK275665
11747	Dog	11 June 2013	Jendouba	Sloul	36.7193985	8.6416702	-	OK275664
12900	Dog	13 June 1014	Jendouba	Ouechtata	36.5028000	8.2497797	-	OK275663
13961	Dog	5 June 2015	El Kef	Eddir	36.2057991	8.7846699	-	OK275666

**Table 2 viruses-13-02473-t002:** List of primers.

Primer Name	Primer Sequence	Sense	Target Gene	Reference
Rab N1	AACACCTCTACAATGGATGCCGACAA	Forward	Nucleoprotein	Nadin Davis S, 1998
Rab N5	GGATTGAC (AG) AAGATCTTGCTCAT	Reverse	Nucleoprotein	Nadin Davis S, 1998
Rab Nfor	TTGT(AG) GA (TC) CAATATGAGTACAA	Forward	Nucleoprotein	Nadin Davis S, 1998
Rab Nrev	CCGGCTCAAACATTCTTCTTA	Reverse	Nucleoprotein	Nadin Davis S, 1998
P for	GAACCATCCCAAAYATG	Forward	Phosphoprotein	Wang L. et al., 2013 [21]
P rev	CTATCTTGCGCAGAAARTTCAT	Reverse	Phosphoprotein	Wang L. et al., 2013

**Table 3 viruses-13-02473-t003:** List of sequences downloaded from GenBank.

Isolate	GenBank Accession Number	Host Species	Location (Country)	Reference
ALG/08-153	GU798549.1	Dog	Algeria	(Talbi et al., 2010)
ALG/08-200	GU798548.1	Human	Algeria	(Talbi et al., 2010)
8676TUN/1986	GU798390.1	Human	Tunisia	(Talbi et al., 2010)
8727TUN/1986	GU798391.1	Human	Tunisia	(Talbi et al., 2010)
86075TUN/1986	GU798389.1	Human	Tunisia	(Talbi et al., 2010)
86130TUN/1986	GU798392.1	Human	Tunisia	(Talbi et al., 2010)
86131TUN/1986	GU798393.1	Human	Tunisia	(Talbi et al., 2010)
86127TUN/1986	GU798395.1	Human	Tunisia	(Talbi et al., 2010)
86128TUN/1986	GU798394.1	Human	Tunisia	(Talbi et al., 2010)
Ariana2	MK981888.1	Dog	Tunisia	[31] (Bonnaud et al., 2019)
DogTN/Bj01	EU643525.1	Dog	Tunisia	(Amouri et al., 2011)
DogTN/Bz98	EU643527.1	Dog	Tunisia	(Amouri et al., 2011)
DogTN/Bz99	EU643528.1	Dog	Tunisia	(Amouri et al., 2011)
DogTN/Jd01	EU643534.1	Dog	Tunisia	(Amouri et al., 2011)
DogTN/Jd96	EU643535.1	Dog	Tunisia	(Amouri et al., 2011)
DogTN/Jd98	EU643536.1	Dog	Tunisia	(Amouri et al., 2011)
DogTN/Ar98	EU643555	Dog	Tunisia	(Amouri et al., 2011)
DogTN/Gb96	EU643532	Dog	Tunisia	(Amouri et al., 2011)
DogTN/Gb98	EU643533	Dog	Tunisia	(Amouri et al., 2011)
DogTN/Jd99	EU643537	Dog	Tunisia	(Amouri et al., 2011)
DogTN/Kf96	EU643540	Dog	Tunisia	(Amouri et al., 2011)
DogTN/Kr02	EU643538	Dog	Tunisia	(Amouri et al., 2011)
DogTN/Kr03	EU643553	Dog	Tunisia	(Amouri et al., 2011)
DogTN/Kr98	EU643539	Dog	Tunisia	(Amouri et al., 2011)
DogTN/Ks00	EU643541	Dog	Tunisia	(Amouri et al., 2011)
DogTN/Ks98	EU643543	Dog	Tunisia	(Amouri et al., 2011)
DogTN/Md95	EU643545	Dog	Tunisia	(Amouri et al., 2011)
DogTN/Md99	EU643546	Dog	Tunisia	(Amouri et al., 2011)
DogTN/Nb00	EU643548	Dog	Tunisia	(Amouri et al., 2011)
DogTN/Ta02	EU643550	Dog	Tunisia	(Amouri et al., 2011)
DogTN/Tn03	EU643554	Dog	Tunisia	(Amouri et al., 2011)
DogTN/Tz96	EU643551	Dog	Tunisia	(Amouri et al., 2011)
DogTN/Zg00	EU643552	Dog	Tunisia	(Amouri et al., 2011)

## Data Availability

Not applicable.

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
