# Peer review of "Molecular Epidemiology of Rabies in Wild Canidae in Tunisia"

_viruses, 2021, doi:10.3390/v13122473_

Round 1

Reviewer 1 Report

The paper has standard methods and extensive genomic analysis of rabies viruses isolated from domestic canine and wild canidae, compared. The results are consistent and conclusions fit well with the lab results and the inferences are fine but the English and expression is poor and needs extensive editing. I cannot do this.

Author Response

Dear reviewer,

We would like to thank you for the revision of the article.

Extensive editing was made on the manuscript regarding English and expressions.

Kind regards

Reviewer 2 Report

The authors undertook an investigation to genetically characterise some rabies viruses and used Bayesian analysis and found that there is no wildlife reservoir in the country. Although the manuscript is generally scientifically sound, I have picked a few issues that the authors should address before it is accepted.

Major:

I. On page 1 - provide full names of the genes when first described - N,M,G,P and L,

ii. There are currently 17 lyssavirus species than the 16 mentioned here,

iii. The authors should also indicate other wildlife reservoirs found in southern Africa, e.g. black-backed jackal species and yellow mongoose (Swanepoel, R., Barnard, B.J.H., Meredith, C.D., Bishop, G.C., Bruchner, G.K., Foggin, C.M. and Hubschle, O.J.B., Onderstepoort Journal of Veterinary Research 60, 323-346, 1993) and bat-eared foxes (Sabeta, C.T., Mansfield, K., McElhinney, L.M., Fooks, A.R. & Nel, L.H. (2007). Molecular epidemiology of rabies in bat-eared foxes (Otocyon megalotis). Virus Research, 129(1): 1-10).

iv. The authors indicated that most of the confirmed rabies cases are from dogs, please provide a %.

v. The authors indicated that the samples sent to the IPT laboratory were tested by the FAT and cell culture- were these done concurrently and what were the e results?

vi. The authors continually use the word isolate throughout the manuscript - were the original  samples actually passages?

Minor:

I. RNA extractions - instead of treated treated use the word homogenised,

ii. Amend to read "extracted RNA was reverse-transcribed"

iii. Under ML - please rewrite :in order to determine taxonomy and clade membership",

iv. Under 2.6 - - reservoir status can be inferred from trends analyses.

rends analyses,

v. Does the placement of dog and wild Canidae viruses infer spillover of infection?

Author Response

Dear Reviewer,

We would like to thank you for your time and efforts in revising the manuscript. Please find here below a response point by point to all of your comments.

Kind regards.

Major:

I. On page 1 - provide full names of the genes when first described - N,M,G,P and L,

The sentence has been corrected in the revised manuscript

ii. There are currently 17 lyssavirus species than the 16 mentioned here,

The sentence has been corrected in the revised manuscript

iii. The authors should also indicate other wildlife reservoirs found in southern Africa, e.g. black-backed jackal species and yellow mongoose (Swanepoel, R., Barnard, B.J.H., Meredith, C.D., Bishop, G.C., Bruchner, G.K., Foggin, C.M. and Hubschle, O.J.B., Onderstepoort Journal of Veterinary Research 60, 323-346, 1993) and bat-eared foxes (Sabeta, C.T., Mansfield, K., McElhinney, L.M., Fooks, A.R. & Nel, L.H. (2007). Molecular epidemiology of rabies in bat-eared foxes (Otocyon megalotis). Virus Research, 129(1): 1-10).

The information has been added to the revised manuscript

iv. The authors indicated that most of the confirmed rabies cases are from dogs, please provide a %.

The sentence has been corrected in the revised manuscript

v. The authors indicated that the samples sent to the IPT laboratory were tested by the FAT and cell culture- were these done concurrently and what were the e results?

We aknowledge that there was an error in our manuscript here. The sentence has been corrected accordingly. In fact, FAT and cell culture (RTCIT) are carried out according to the OIE recommendations which are ; FAT is undertaken on all samples received while RTCIT is carried only on negative samples for confirmation. All samples considered in this study were found positive to FAT, and thus have not been analyzed later by RTCIT.

vi. The authors continually use the word isolate throughout the manuscript - were the original  samples actually passages?

All the samples used in this study are original brain samples from taken from the animals during necropsy and stored at -70°C.

Minor:

I. RNA extractions - instead of treated treated use the word homogenised,

The sentence has been corrected in the updated manuscript

ii. Amend to read "extracted RNA was reverse-transcribed"

The sentence has been corrected in the updated manuscript

iii. Under ML - please rewrite :in order to determine taxonomy and clade membership",

The sentence has been corrected in the updated manuscript

iv. Under 2.6 - - reservoir status can be inferred from trends analyses.

rends analyses,

Thank you for the suggestion.

v. Does the placement of dog and wild Canidae viruses infer spillover of infection?

In our study, we used multiple analyses to confirm the absence of spillover in wild life ; taxonomy, clade membership, gene sequence similarity and timescale Bayesian phylogeny. Spatial distribution of cases was not used as an argument itself, but as a verification of the hypothesis.